# Detecting the spin-polarization of edge states in graphene nanoribbons

Jens Brede [1,2], Nestor Merino-Díez [1,2], Alejandro Berdonces-Layunta [1,2], Sofía Sanz [1], Amelia Domínguez-Celorrio [3], Jorge Lobo-Checa [3,4,5], Manuel Vilas-Varela [6], Diego Peña [6], Thomas Frederiksen [1,7], José I. Pascual [7,8] ✉, Dimas G. de Oteyza [1,2,9] ✉ & David Serrate [3,4,5] ✉

Low dimensional carbon-based materials can show intrinsic magnetism associated to p-electrons in open-shell π-conjugated systems. Chemical design provides atomically precise control of the π-electron cloud, which makes them promising for nanoscale magnetic devices. However, direct verification of their spatially resolved spin-moment remains elusive. Here, we report the spin-polarization of chiral graphene nanoribbons (one-dimensional strips of graphene with alternating zig-zag and arm-chair boundaries), obtained by means of spin-polarized scanning tunnelling microscopy. We extract the energy-dependent spin-moment distribution of spatially extended edge states with π-orbital character, thus beyond localized magnetic moments at radical or defective carbon sites. Guided by mean-field Hubbard calculations, we demonstrate that electron correlations are responsible for the spin-splitting of the electronic structure. Our versatile platform utilizes a ferromagnetic substrate that stabilizes the organic magnetic moments against thermal and quantum fluctuations, while being fully compatible with on-surface synthesis of the rapidly growing class of nanographenes.

Conventional magnetism is based on the spin of unpaired d- or f-shell electrons in the outermost atomic orbitals. In contrast, in certain organic structures, carbon p-electrons can form π-orbitals with a net magnetic moment[1,2]. The vision of exploiting π-orbital magnetism in applications involving spin-polarized currents and spin-based quantum information[1–5] is inspired by two distinct properties: weak spin-orbit and hyperfine couplings (two of the main channels responsible for the relaxation and decoherence of electron spins)[1–3,6,7], and delocalization in π-orbitals with high spin-wave stiffness[1,2,8]. However, the experimental realization of the associated open-shell molecular structures is challenging. Therefore, it is only recently that they are

becoming accessible by on-surface synthesis under vacuum conditions[9], rendering their characterization of utmost interest.

Within the wide range of carbon nanostructures predicted to display π-magnetism[9], graphene nanoribbons (GNRs) are among the most interesting for potential applications due to their intrinsic length[10,11], which facilitates contacting and integration into device structures[12,13]. Aside from the presence of vacancies or heteroatoms, for GNRs to exhibit magnetic properties they must display zigzag edges, either continuously throughout the ribbon (zGNRs) or in periodic alternation with armchair segments (chiral, chGNRs)[10,14–16]. In both cases, the ribbons develop edge states that decay exponentially

[1]Donostia International Physics Center, San Sebastián E-20018, Spain. [2]Centro de Física de Materiales (MPC), CSIC-UPV/EHU, San Sebastián E-20018, Spain. [3]Instituto de Nanociencia y Materiales de Aragón (INMA), CSIC-Universidad de Zaragoza, Zaragoza E-50009, Spain. [4]Departamento de Física de la Materia Condensada, Universidad de Zaragoza, Zaragoza E-50009, Spain. [5]Laboratorio de Microscopias Avanzadas (LMA), Universidad de Zaragoza, Zaragoza E-50009, Spain. [6]Centro Singular de Investigación en Química Biolóxica e Materiais Moleculares (CiQUS) y Departamento de Química Orgánica, Universidade de Santiago de Compostela, Santiago de Compostela E-15782, Spain. [7]Ikerbasque, Basque Foundation for Science, Bilbao E-48013, Spain. [8]CIC nanoGUNE BRTA, San Sebastián E-20018, Spain. [9]Nanomaterials and Nanotechnology Research Center (CINN), CSIC-UNIOVI-PA, El Entrego E-33940, Spain. ✉e-mail: ji.pascual@nanogune.eu; d.g.oteyza@cinn.es; serrate@unizar.es

towards the ribbon's interior. These electronic states are predicted to be spin-polarized[10,15,16]. zGNRs and chGNRs have been synthesized with atomic precision on Au(111) substrates[17–20]. The presence of the edge states has been confirmed in both cases[17,20], but a direct experimental proof of their spin polarization is still lacking. In the larger class of open-shell carbon nanostructures, spatially resolved magnetic signals have only been observed in a few cases involving magnetic field-dependent Zeeman splittings[21–23]. Otherwise, only indirect hints of the magnetism have been obtained, either by an analysis of the frontier orbitals' density of states[24,25], Kondo resonances[21,22,26], inelastic spin-flip excitations[23,26,27] or Coulomb gaps[17,28]. In any case, the detection of an intrinsic remanent spin-polarization of π-orbitals has never been obtained to date for any carbon-based material. Actually, the weak spin-orbit coupling[3], a central advantage of carbon-based magnetism, imposes at the same time the main drawback to resolve a stationary spin moment in these systems: the practically zero magnetic anisotropy of $sp^2$ carbon atoms.

To circumvent this constraint, we utilize a ferromagnetic $GdAu_2$ monolayer on Au(111) as a substrate to stabilize the spin-polarization of chGNR atop it[29]. We obtain chGNRs with edges oriented along the chiral (3,1) graphene lattice vector[15] and $c = 8$ carbon atoms across their width (thus (3,1,8)-chGNRs) by deposition and appropriate thermal treatments (see Methods) of the reactant 2″,3′-dibromo-9,9′:10′,9″:10″,9‴-quateranthracene[20] (DBQA). Subsequent characterization by spin-polarized scanning tunneling microscopy and spectroscopy (SP-STM/STS), supported by mean-field Hubbard (MFH) model calculations, unravel the spatially and energetically resolved spin-polarization of the ribbon's frontier states.

For the characterization of an unexplored magnetic ground state by means of SP-STM, it is convenient to arrange the sample under study in coexistence with a surface that has a well-known spin-resolved electronic structure. A $GdAu_2$ monolayer on Au(111)[29] is an excellent

candidate. Its ability to catalyze nanographene polymerization via Ullmann coupling[30] has already been proven[31,32] and, at the same time, it orders ferromagnetically below 19 K[33] with a large easy-plane magnetic anisotropy[34], thereby showing strong in-plane contrast in SP-STM measurements[35].

We achieve long (3,1,8)-chGNRs from DBQA precursor molecules (Supplementary Fig. S1) on $GdAu_2$ using a very similar procedure as the one previously reported in the case of Au(111)[20] (see Supplementary Experimental Methods). Figure 1a shows an overview image of (3,1,8)-chGNRs on $GdAu_2$. The moiré superlattice caused by the superposition of the $GdAu_2$ lattice (hexagonal unit cell of $5.41 \pm 0.03$ Å) and the underlying Au(111) lattice[29,36] is clearly visible. In the $GdAu_2$ lattice, each Gd atom is sixfold coordinated with Au atoms, which appear in STM images as dark and bright spots, respectively (Fig. 1b). The edges of (3,1,8)-chGNRs are composed of alternating zigzag and armchair segments (see molecular scheme in Fig. 1b). The GNRs grow preferentially with their longitudinal axis parallel to high symmetry directions of the Gd atomic lattice, either the [1$\bar{1}$0] (e.g. the central ribbon in Fig. 1b) or the [11$\bar{2}$] directions of the Au(111) substrate (see also Supp. Fig. S2)).

In addition to enabling the formation of high-quality (3,1,8)-chGNRs, the $GdAu_2$ surface exhibits in-plane magnetic domains, as revealed in the spin-resolved $dI/dV$ maps at $V_b = 3$ V shown in Fig. 1c, d, taken with bulk Cr-tips (further details at Supplementary Fig. S3)[35]. We choose bulk Cr tips for the SP-STM because they are well-suited to probe magnetic contrast in soft ferromagnets like $GdAu_2$ thanks to their large switching fields of several Tesla[37] and negligible stray fields[38]. Crystallographic antiphase domain boundaries (APB) provide an atomically sharp boundary between $GdAu_2$ domains with opposite magnetization direction (Fig. 1e, f). This serves to control that the tip's spin sensitivity remains unaltered throughout the whole measurement. We select target GNRs located in a magnetic domain with homogeneous magnetization and close to an APB. Ramping the

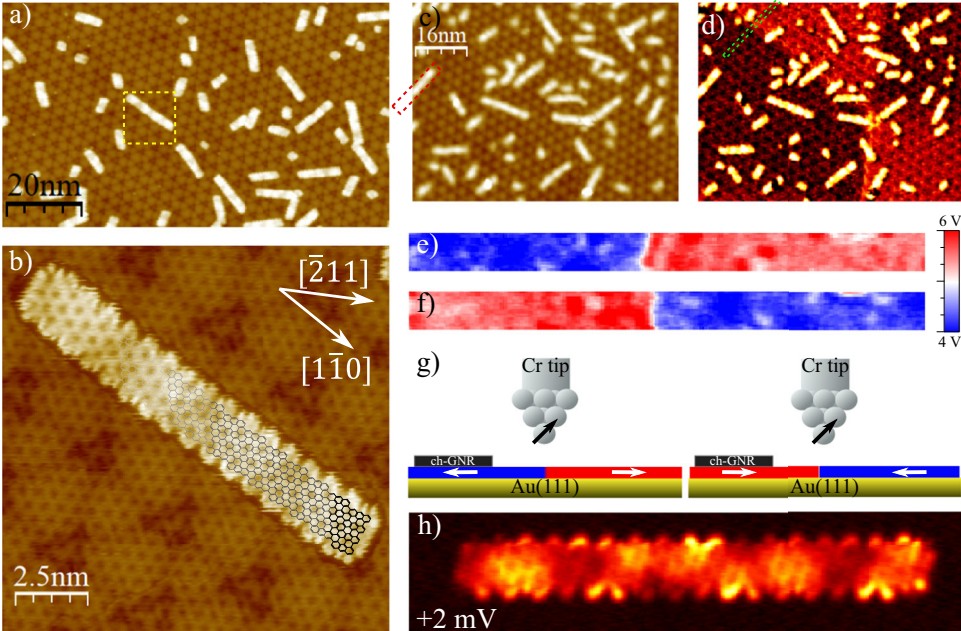

**Fig. 1 | Experimental set-up for the magnetic characterization of (3,1,8)-chGNRs on GdAu₂. a** Survey of cGNRs on $GdAu_2$ (SP: -1.5V, 100 pA). **b** Zoom of the area within the yellow square in (**a**), STM topography mixed with the double derivative image to enhance the Gd lattice (SP: 1V, 50pA; W tip). **c, d** Simultaneous topography image and $dI/dV$ spin-polarized map with in-plane sensitive Cr tip (SP: 3V, 50 pA; $V_{mod} = 20$ mV rms; $B = 0$ Tesla). The image contains structural antiphase boundaries (APB) which induce antiferromagnetic coupling among neighbouring domains. **e, f** Zoom of the region enclosed by the green rectangle in (**d**) of two different

remanent magnetic states ($B = 0$ T) of the substrate obtained after cycling the field at maximum positive and negative out-of-plane field strength of $\pm 3$ Tesla. This is an example of how we control the magnetic state of the tip-sample system for the subsequent magnetic characterization of the ribbons. **g** Cartoon model of the entire tip-sample system in (**e**) and (**f**) where arrows represent the local magnetic moment. **h** Constant height $dI/dV$ map at 2 mV ($V_{mod} = 0.5$ mV rms) of the $N = 17$ ribbon marked in (**c**) with the same Cr tip (feedback opened at ribbon centre with SP: 50 mV, 300 pA).

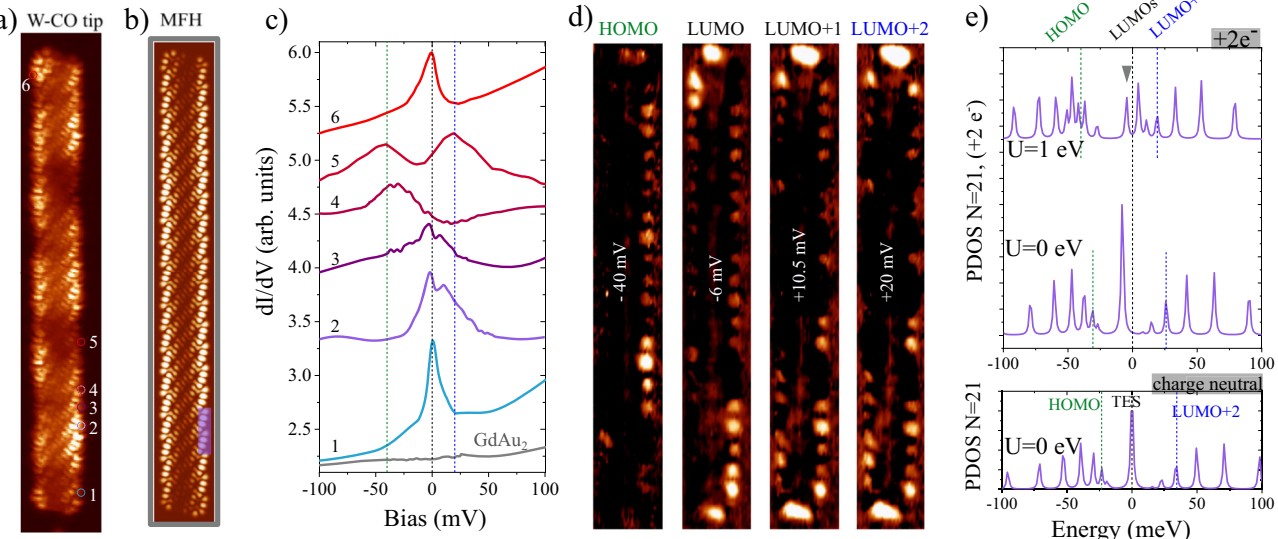

**Fig. 2 | Spin averaged electronic structure of (3,1,8)-chGNRs on GdAu₂.** ($T = 4.3$K, $B = 0$ T, W tip). **a** Constant height current with CO functionalized tip at $V_b = -3.7$ mV of a $N = 21$ ribbon (open feedback at ribbon's centre with current SP of 50 pA). **b** Simulation of the GNR DoS within the Mean Field Hubbard (MFH) model at the energy indicated by the grey arrow in (**e**), $U = 1$ eV and two electrons added (see Methods). **c** High-resolution $dI/dV$ spectra (SP:−100 mV, 200 pA, $V_{mod} = 0.7$ mV) taken at the positions given by the colored circles in (**a**). **d** Constant current $dI/dV$ maps at the energy values marked by dashed lines in (**c**) with the same color code

(SP: -150 mV, 125pA, $V_{mod} = 10$ mV). **e** Projected DoS from MFH calculations (see Theoretical Methods) at the atomic sites enclosed by the transparent rectangle in (**b**). Bottom panel corresponds to the charge-neutral case without e-e interactions, top panel corresponds to the charged system with 2 electrons for $U = 0$ and $U = 1$ eV. Dashed lines in (**c**) and (**e**) indicate the energy positions of the HOMO (green), LUMO (black) and LUMO+2 (blue) named after their equivalent states in the charge-neutral specimen, where the zero-energy peak corresponds to the topological end-state (TES, see main text).

external out-of-plane field to ±3 Tesla and back to zero allows us to repeatedly switch the remanent state of the substrate underneath the target GNR (see Supplementary Note 2), as shown in Fig. 1d–f and sketched in Fig. 1g. Supplemental Fig. S9 demonstrates that this switching is caused by the domain wall displacement in the regions at either side of the APB, which is in turn driven by the in-plane projection of the external field, arising from the practical difficulty to mount the sample with its normal direction perfectly aligned with the vertical axis. In the following, we refer to magnetic states with high and low differential conductance at 3 V as parallel (P) and antiparallel (AP) states to the spin direction of the tip.

After setting the magnetic state of the substrate, we proceed with the characterization of the target GNR (Fig. 1h, the characterization region is marked in Fig. 1c, lying on the left domain of Fig. 1e-f). The density of states (DoS) around the Fermi level can beretrieved from low-bias $dI/dV$ maps. The result for a ribbon with $N = 17$ precursor units (Fig. 1h), measured with metallic tips (Cr or W), is representative of the DoS obtained for other GNRs with lengths varying from 10 to 23 precursor units. The internal structure observed in the central region corresponds to the GdAu₂ lattice, and the moiré contrast is visible through the ribbon.

Importantly, we resolve the predicted high DoS at the ribbons's edges[15,16,20]. The well-known short decay length of edge states into the vacuum[39] suppresses their characteristic intensity in constant height DoS images taken with metallic tips (Supplementary Figs. S6 and S8). However, the expected DoS distribution is recovered when using CO-functionalized W-tips, as shown in Fig. 2a for a $N = 21$ chGNR, including the structure of the wave function extending into the central part of the ribbon. The pronounced edge states are, in most cases, convoluted with electronic contrast due to the moiré pattern underneath (See Supplementary Note 1), rather than displaying pure quantum well states caused by the finite length[19,40].

We cannot resort to CO functionalization for SP-STM experiments, which require metallic tips. To probe the molecular states with this kind of tips it is necessary to acquire grids of $dI/dV$ spectra over the chGNR regulating the current at each pixel (see Methods). Selected

point spectra taken with a metallic W tip are shown in Fig. 2c, exhibiting clear peaks between -6 mV and 28 mV, and a first fully occupied state at −40 mV. Figure 2d shows $dI/dV$ slices at constant $V_b$ for these states, where a certain repetition period along the edge can be discerned. Although the influence of the moiré is lower than in constant height scans (Fig. 2a), it still coexists with the characteristic periodicities of the quantum well states (Supplementary Note 1). Both contributions can be separated by taking the fast Fourier transform (FFT) of the conductance. In particular, the moiré spatial pattern is energy independent, while from the FFT analysis, a characteristic dispersive behavior can be extracted for the wave vectors of each molecular state[19, 39](See Supplementary Fig. S6). This analysis permits to assign the spectroscopic resonances to discretized states emerging from the conduction and valence bands of the infinitely long GNRs.

The peak at -40 mV arises from the valence band, and thus corresponds to the Highest Occupied Molecular Orbital (HOMO) of the charge-neutral ribbon. The peaks crossing the Fermi level ($V_b$ -0 mV) and above ($V_b$ -20 mV) relate to the conduction band and are the lowest unoccupied molecular orbitals (LUMO, LUMO+1,...) of the charge-neutral ribbon. The appearance of the LUMO right at the Fermi level on GdAu₂/Au(111) is a distinct feature of charge transfer from the substrate to the GNR. This is a consequence of the reduced work function of GdAu₂/Au(111) with respect to Au(111)[32], which promotes a slight electron doping of the GNR and drives the LUMO states stemming from the bottom of the conduction band below the Fermi level.

The LUMO peak centred at zero bias exhibit an additional splitting of approximately 12 mV across the Fermi level at some locations (e.g. curves 2 and 3 in Fig. 2c). This sub-structure is observed whenever a sizable DoS centred at zero bias is present at chiral edges. To elucidate the origin of the edge electronic structure, we solved the tight-binding Hamiltonian for the π-electron system, including a MFH term to account for electronic correlations (see Methods). For non-interacting electrons ($U = 0$), the addition of two electrons to the charge-neutral ribbon shifts the simulated DoS spectra at the chiral edge by about −8 meV, (Fig. 2e). The pristine LUMO+2 at 34 meV sits now at 26 meV, and the zero-energy end state of topological origin[20] (TES) located at the

termini shifts to -8 meV. We introduce electron-electron (e-e) interactions via an on-site Coulomb repulsion term $U \neq 0$. With $U = 1$ eV we are able to reproduce our experimental $dI/dV$ spectra (top curve in Fig. 2e). First, the main intensity of the first fully occupied states clusters around -40 meV. Second, the structure around the Fermi level splits into two peaks separated by 9 meV, very close to our experimentally determined faint gap (Fig. 2c). This feature arises as a consequence of the Coulomb repulsion. The TES splits at $U = 0.8$ eV into a singly occupied and a singly unoccupied orbital, while the low-energy LUMOs are shifted to more negative energies, mixing with each other in the range of $U$ -1 eV (see Supplemental Fig. S4). Figure 2b shows the theoretical LDoS distribution of the occupied state marked by a grey arrow in Fig. 2e, which is in excellent agreement with the experimental DoS in Fig. 2a.

Localized one-dimensional (1-D) states at the zigzag edges of graphene nanostructures[41] have been predicted to become spin-polarized by MFH models[15,16] and first-principles calculations[42–46]. The driving mechanism is the energy gain of the system obtained by depleting a spin-degenerate doubly occupied state near the Fermi level, for which the Coulomb repulsion would otherwise introduce a much higher internal energy. This phenomenon is prone to occur in localized electronic states, because the Coulomb repulsion energy grows as electrons get confined in a more reduced space.

Although our (3,1,8)-chGNRs do not have pure zigzag edges (see Fig. 1b), its periodic zigzag segments are known to retain intense localized edge states stemming from flat electronic bands near the Γ point of the 1-D Brillouin zone (see Fig. 2a and d, and Supplemental Fig. S6b)[15,20,46,47]. If the two edges are far enough as to be considered independent, the edge states are expected to be metallic in (3,1,$c$)-GNRs[15]. However, in narrower ribbons, the coupling between both edge states can open a hybridization gap[15,19,20]. In the case of $c = 8$, this gap amounts to approximately 20 meV, and is centred at the Fermi level[20]. On (3,1,8)-chGNRs/GdAu₂, the slight electron doping causes an increase of the chemical potential in the ribbon, and instead of a gap at the Fermi level we find a partially filled state, at -35 mV above the HOMO (see Fig. 2c). This state is still close to the conduction band minimum (Supplementary Fig. S6), and therefore it will display a similar degree of localization at the edge as in the metallic case of wider ribbons ($c > 8$)[15,46,47]. This is illustrated in the aforementioned constant height scans with CO-tips (Fig. 2a and Supplemental Figs. S6 and S8) or in grids with current regulation for each pixel (Fig. 2d). Thus, they are excellent candidates to display magnetic instabilities associated to e-e correlations.

The splitting induced by e-e interactions endows different spin quantum numbers to the occupied/unoccupied states. From this, it follows that an inversion of the spin polarization must necessarily exist[15,16,42]. In the following, we provide evidence of such sign inversion in the spin polarization, setting the experimental hallmark for itinerant magnetism in edge states of nanographenes.

Figure 3a shows a $N = 15$ chGNR, scanned under constant-height conditions for several magnetic $P$ or $AP$ states of the underlying GdAu₂ with homogeneous magnetization in the characterization region (see Supplementary Fig. S9 for a description of the magnetic history). The spin asymmetry in Fig. 3b, calculated from the $dI/dV$ maps at the energies of interest as $S_a = 100 \times [(dI/dV)_{AP} - (dI/dV)_P]/[(dI/dV)_{AP} + (dI/dV)_P]$, is proportional to the spin polarization of the system right at the measurement energy. Figure 3d shows spectral lines along the chiral edges of the spin averaged DoS, i.e. $[dI/dV_{AP} + dI/dV_P]/2$. Here, the previously discussed splitting for $N = 21$ (theoretical spectra for $N = 15$ available at Supplementary Fig. S12d) manifests again in the regions of larger DoS as two peaks at $V_b = -7$ mV and at $V_b = +5$ mV, noticeable at the bottom of the left edge (region 1 in Fig. 3a) and at the centre of the right edge (region 3 in Fig. 3a).

Regions 1 and 3 are also the positions with clear spin contrast in Fig. 3b. Furthermore, we find experimentally a change of sign in the

spin polarization across the Fermi level of ±8%, in neat agreement with the predicted magnetic state driven by e-e correlations. Therefore, the 12 mV gap observed around the Fermi level (Figs. 2c and 3d) can be safely attributed to a spin splitting that emerges to accommodate e-e correlations in the partially occupied LUMO edge state. If these two peaks were two different quantum-well states of the conduction band magnetized by the proximity with the substrate, they would not have a different spin polarization than the inner region of the ribbon, and their spin asymmetry would not change sign across the Fermi level. Note that the spin polarization of the underlying GdAu₂ obtained with the same kind of bulk Cr-tips is very small (<4%), and energy-independent in this bias regime (Supplementary Fig. S3).

The energy dependence of the spin polarization is further highlighted in single point $dI/dV$ spectra (Fig. 3e). As for the constant height images (Fig. 3a), the spin asymmetry is obtained as $S_a(eV_b) = (AP - P)/(AP + P)$, and represented by the green curves. All positions other than the ribbon periphery are characterized by featureless spectra whose spin asymmetry between $AP$ and $P$ states is below our experimental confidence ($\leq 1.5\%$). Region 1 and 3 exhibit the canonical behaviour discussed above for a correlations splitting, with $S_a = +7\%$ at $V_b = -10$ mV and $S_a = -6\%$ at $V_b = +6$ mV. Region 2, with a much lower intensity of the edge state (see Fig. 3d), only displays a small signal of $S_a = -3.3\%$ at $V_b = +4$ mV, barely above our experimental uncertainty, and $S_a \simeq 0$ at the energy of the negative bias peak.

Figure 3c displays the spin polarization obtained from MFH calculations with the set of parameters determined earlier ($U = 1$ eV, 2 electrons added, see Methods), which is in good qualitative agreement with the experimental results (see Fig. 3b). This comparison allows us to understand the detected pattern of the spin polarization. On the one hand, $U = 1$ eV is smaller than the expected theoretical value for free-standing nanoribbons, namely around 2-3 eV[44,48,49] which in our case is justified by the hybridization of the GNR $p_z$ orbitals with the metallic substrate. This causes a spread of the electronic wave function and facilitates the charge screening, both detrimental to the energy scale of e-e Coulomb interactions. Consequently, the correlations splitting (-10 meV) is smaller than the FWHM of the molecular orbitals (-30 meV, see Fig. 2c), leading to a weakening of the spin polarization.

On the other hand, the spin polarization along the edge is not homogeneous, contrary to the expectation for the ground state of isolated chGNRs (see Supplementary Fig. S11a). Our model calculation, shown in Fig. 3c, clearly captures the oscillatory pattern, indicating that it is intrinsic to the electronic structure. For the electron-doped chGNR, the energy spectrum of the ribbon downshifts gradually with increasing doping and $U$ (Fig. 2e). The effect of the $U$ term consists in mixing the first three unoccupied quantum well states of the non-interacting case, whose energies are concomitantly corrected and shifted towards the Fermi level (see Supplemental Fig. S4). As a result, the spin-split molecular orbital formed at Fermi level (see Fig. 2e for $N = 21$ and Supplementary Fig. S11d for $N = 15$) is a hybrid of the TES[20] and the low-order quantum well states, each one of them with its own characteristic pattern of ridges and nodal planes. Consequently, the edge spin polarization displays oscillations (Fig. 3b, c). This hybridization is caused by the loss of particle-hole symmetry introduced by the electron doping (see Supplementary Fig. S11). It is also responsible for the difference in the spatial distribution of the spin densities above and below the Fermi level (Fig. 3b and c, see Supplementary Fig. S12 for higher energies). Furthermore, in the calculation, the two added electrons are evenly distributed among the two spin subspaces (see Methods), which in combination with the inversion symmetry of the system, causes the total magnetic moment to be zero for all states, as reflected in the mirrored spin distributions in opposite edges. This is not happening in the experimental data, since integer charge transfer is an approximation that the real metal-supported GNRs will certainly not meet.

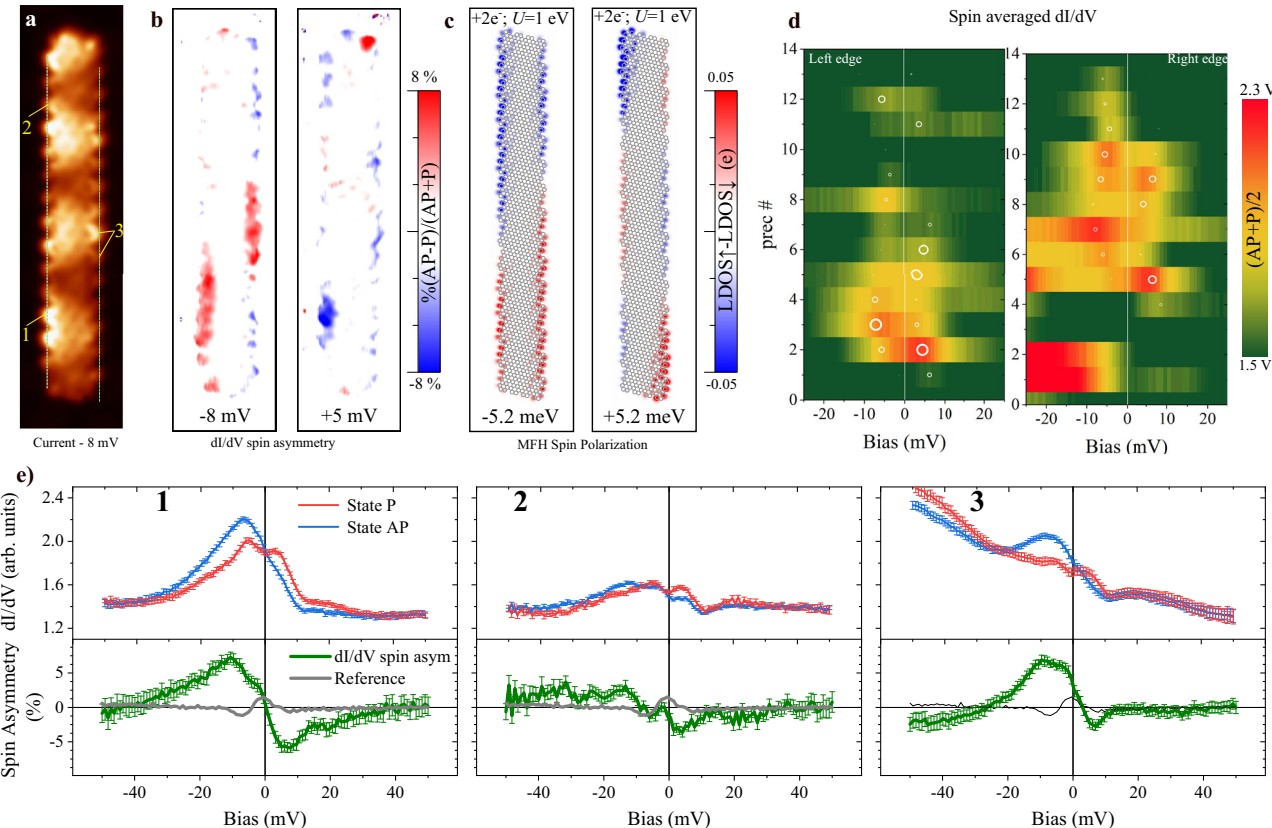

**Fig. 3 | Spin polarized edge states in chiral graphene nanoribbons.** ($T$ = 1.2 K, in-plane sensitive bulk Cr-tip, $N$ = 15 precursors). Control tests of the obtained magnetic contrast are provided at Supplementary Note 2. All images are 3.9 × 15.8 nm² and taken at constant height after opening the feedback over the ribbon's centre (SP: 20 mV and 100 pA, $V_{mod}$ = 0.5 mV). $dI/dV$ point spectroscopies taken with set point of 100 mV and 250 pA ($V_{mod}$ = 0.5 mV). **a** Tunneling current image of the ribbon at $V_b = -8$ mV. **b** $dI/dV$ spin asymmetry calculated from the constant height differential conductance images as %($AP - P$)/($AP + P$) at $V_b = -8$ mV (left) and $V_b = +5$ mV (right). Note the sign change of the spin polarization when crossing the Fermi level. **c** Spin polarization of the GNR at eigenergies indicated in the panels, obtained with $U$ = 1 eV from the MFH model (see Methods). The spin polarization is spanned in a real space grid and the plot is obtained by slicing 3.5 Å above the molecular plane. **d** Stack plot of the spin averaged point spectroscopies, ($AP + P$)/2, taken along the green dashed lines in (**a**). The position/diameter of the white circles represent the centre/area of Lorentzian functions used to fit each individual spectrum together with a baseline. **e** Spin resolved point spectroscopy at the positions indicated by the yellow numbers in (**a**) for the $P$ (red curves) and $AP$ (blue curves) states. The resulting energy-resolved spin polarization is given by the green curves in the bottom panel of each graph. Light grey lines represent the background instrumental asymmetry obtained over GdAu₂ or the ribbon centre (both are identical). Error bars are the standard deviation with confidence of 68 % obtained from 20 individual curves measured at each point.

There are also extrinsic factors that can justify the quantitative deviation of the edge magnetization with respect to the theoretical simulation of Fig. 3c. These are related to local perturbations of the interaction with the substrate, like for instance the charge accumulation or depletion induced by the local surface potential (Supplementary Note 1 and Figure S5). Another possible factor is the magnetic exchange with the substrate, which is expected to fluctuate within the GNR extent for each particular relative stacking of the C and Gd lattices (among the various adsorption geometries reported in Supplementary Figs. S2, S6 and S8). Nevertheless, for chGNRs over homogeneously magnetized GdAu₂ regions, neither of these extrinsic factors prevent us in practice from resolving the intrinsic nature of the detected spin polarization.

The magnetic interaction with the substrate plays an important role though. It provides a defined quantization axis for the chGNRs magnetic moment, as well as a finite mean field exchange interaction which stabilizes the magnetic moment of the edge states against quantum and thermal fluctuations. To corroborate this idea, and using ribbons over GdAu₂ regions with inhomogeneous magnetization, we demonstrate the independent switching of the spin direction in a portion of the ribbon while the rest remains unchanged (Supplementary Fig. S13 and Supplementary Note 3). This is necessarily caused by a local exchange coupling to the GdAu₂ magnetization. This interaction

does not induce a magnetic moment in the chGNR, but it does allow access to the stationary edge's spin polarization by lifting the degeneracy of spin states with opposite direction.

Altogether, by synthesizing chiral graphene nanoribbons on top of a magnetic GdAu₂ monolayer, we have been able to access with exquisite spatial and energy resolution the spin-polarization of its edge states by SP-STM/STS. Doing so, a long-standing challenge has been resolved that not only reveals important differences with respect to the originally expected spin polarization of charge-neutral ribbons, but also sets the stage to similarly characterize the immense amount of magnetic carbon-based materials that are being synthesized lately.

## Methods
### STS/STM measurements
All measurements have been performed at the SPECS-JT-STM of the Laboratory for Advanced Microscopy (University of Zaragoza), which features a base temperature of 1.17 K and an out-of-plane magnetic field up to 3 Tesla provided by a dry superconducting split-coil. The whole facility operates under ultra-high-vacuum conditions ($P$ -1 × 10⁻¹⁰ mbar). The tip is grounded and the tunneling bias $V_b$ is applied to the sample. Data has been taken at $T$ = 1.2 K unless stated otherwise. Differential tunneling conductance $dI/dV$ is acquired using a lock-in amplifier at a frequency of 933 Hz and r.m.s. modulation given by $V_{mod}$.

STM images and $dI/dV$ maps were taken either in constant height or in constant current mode, using a regulation distance determined by the set point (SP) indicated at the corresponding caption for each data set. In the case of the constant current $dI/dV$ maps as those shown in Fig. 2d of the main text and in Supplementary Fig. S6b, the image is formed by slicing at fixed $V_b$ the $dI/dV$ signal from a dense grid where a full spectra is acquired ramping $V_b$ at each pixel. Spin-polarized STM (SP-STM) has been carried out using bulk Cr tips. Tips are prepared by electrochemical etching of elongated pure Cr flakes and subsequent field emission cleaning (120 V, 1 $\mu$A, 1 hour) at the STM head. Then they are submitted to positive voltage pulses until the expected in-plane spin contrast of 20% or more at $V_b = 3$ V is obtained. We avoid tip-sample indentation in the whole process, and discard tips with unstable in-plane contrast at zero field or abrupt changes of their spin polarization during field sweep. CO functionalization of Cr and W tips is achieved by scanning a line across the adsorbate with feedback on and low gap resistance determined by a set point ranging $V_b = 2 - 20$ mV and $2 - 5$ nA (for W tips we obtain a better success rate of 9/10 for $V_b < 0$).

## Sample preparation

The Au(111) single crystal from Mateck GmbH was cleaned by repeated Argon sputtering and annealing processes at 510 °C. GdAu$_2$ alloy is grown on the clean Au(111) surface by sublimating Gd using an e-beam source at a rate of approximately 0.5 ML (referred to the Au(111) lattice) in 9 minutes while the substrate is held at 320 °C. The reactant 2″,3′-dibromo-9,9′:10′,9″:10″,9‴-quateranthracene (DBQA) is then deposited on GdAu$_2$ from a homemade resistive evaporator. Subsequent on-surface synthesis of (3,1,8)-chGNRs takes place in a two steps reaction, which is detailed in Supplementary Fig. S1.

## Mean Field Hubbard calculations

To theoretically describe GNRs and their spin physics we employ the Hubbard Hamiltonian within the mean-field approximation (MFH)[50],

$$H = - t_1 \sum_{\langle i,j \rangle, \sigma} c_{i\sigma}^\dagger c_{j\sigma} - t_2 \sum_{\langle\langle i,j \rangle\rangle, \sigma} c_{i\sigma}^\dagger c_{j\sigma} - t_3 \sum_{\langle\langle\langle i,j \rangle\rangle\rangle, \sigma} c_{i\sigma}^\dagger c_{j\sigma} \\ + U \sum_i \left( n_{i\uparrow} \langle n_{i\downarrow} \rangle + \langle n_{i\uparrow} \rangle n_{i\downarrow} - \langle n_{i\uparrow} \rangle \langle n_{i\downarrow} \rangle \right) \quad (1)$$

where $c_{i\sigma}$ ($c_{i\sigma}^\dagger$) is the annihilation (creation) operator for an electron with spin $\sigma$ at site $i$, and $n_{i\sigma} = c_{i\sigma}^\dagger c_{i\sigma}$ is the number operator. $t_{1,2,3}$ are the hopping terms corresponding to the interaction between first, second and third-nearest neighbors (3NN), respectively. For these parameters we use the numerical values $t_1 = 2.7$, $t_2 = 0.2$, $t_3 = 0.18$ eV[44], corresponding to neighbor distances between $d_1 < 1.6$Å $< d_2 < 2.6$Å $< d_3 < 3.1$Å, respectively. For the Coulomb repulsion term, we use $U = 1$ eV. The reason for using this 'small' value compared to other related works[21,26], is because we see evidence of hybridization between the samples and the substrate (see main text). Numerically, we solve the Schrödinger equation for Eq. (1) using our custom-implemented Python package HUBBARD[51]. Here, the average number operators $\langle n_{i\sigma} \rangle = \sum_n f_{\sigma n} |b_{i\sigma}^n|^2$ for each spin component $\sigma = \{\uparrow, \downarrow\}$, are calculated by summing the eigenvectors (of coefficients $b_{i\sigma}^n$) resulting from the diagonalization of the MFH Hamiltonian, up to the last occupied $n$th molecular orbital (which depends on the present number of electrons), as encoded in the Fermi function $f_{n\sigma}$.

We compute the LDOS as,

$$\text{LDOS}(\mathbf{r}, E, \sigma) = \sum_n |\Psi_{n\sigma}(\mathbf{r})|_{n\sigma}^2 \frac{\gamma/\pi}{(E_{n\sigma} - E)^2 + \gamma^2}, \quad (2)$$

where

$$\Psi_{n\sigma}(\mathbf{r}) = \sum_i b_{i\sigma}^n \, \phi(\mathbf{r} - \mathbf{R}_i) \quad (3)$$

denotes the individual orbital contribution corresponding to the wave function coefficient $b_{i\sigma}^n$ for state $n$ at site $i$ and spin orientation $\sigma$. The carbon $2p_z$ basis orbital $\phi(\mathbf{r} - \mathbf{R}_i)$, centered at position $\mathbf{R}_i$, is chosen to be described by the hydrogen-like atomic orbital with effective core charge $Z_{\text{eff}} = 3.2$[52]. We note that while we consider an orthogonal TB description, the orbitals that we used to describe the real space wave function overlap at the carbon-carbon bond distances (1.42 Å). However, the qualitative picture should not be affected by this, as we only consider the real space expansion for visualization purposes. The Lorentzian broadening, set to $\gamma = 1$ meV, is introduced to account for the mixing of energetically closely spaced orbitals. In all cases we slice the grid at a height of $z = 3.5$ Å above the chGNR. We plot these quantities using our HUBBARD python package[51]. The spin polarization at an energy $E$ can be obtained in a similar way,

$$\text{Polarization}(\mathbf{r}, E) = \sum_{\sigma = \pm 1} \sigma \text{LDOS}(\mathbf{r}, E, \sigma). \quad (4)$$

The spin polarization can be also integrated over an energy window, where the total spin polarization would correspond to the integration among all occupied states,

$$\text{Polarization}(\mathbf{r}) = \int_{E_{\min}}^{E_{\max}} \text{Polarization}(\mathbf{r}, E) dE, \quad (5)$$

We charge the system by adding one electron with spin up and one electron with spin down (total magnetization $S_z = 0$), since according to our calculations this is the state of lowest energy. The state $S_z = 1$ is 3 meV above the ground state.

## Data availability

Raw data and processed datasets generated in this study are available at the public repository DIGITAL.CSIC (http://hdl.handle.net/10261/336588) under a permanent D.O.I.

## Code availability

Computer code used to solve the Hubbard Hamiltonian and generate the projected density of states and images of the real space spin polarization is available at reference[51] under https://doi.org/10.5281/zenodo.4748765.

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

## Acknowledgements

We acknowledge financial support from the Spanish Ministry of Science and Innovation MICIN through grant nos. PID2019-107338RB-C64 (D.S. and J.L.C.), PID2019-107338RB-C61 (J.I.P.), PID2019-107338RB-C62 (D.P.), PID2019-107338RB-C63 (D.G.O.), PID2020-115406GB-I00 (T.F.) funded by AEI/10.13039/501100011033; grant no. PCI2019-111933-2; and red temática RED2018-102833-T. This work was also supported by European Regional Development Fund (ERDF) under the program Interreg V-A España-Francia-Andorra (grant no. EFA194/16 TNSI), the European Union (EU) H2020 program through the FET-Open project SPRING (Grant Agreement No. 863098, J.I.P. and T.F.) and the NextGeneration/PRTR grant no. TED2021-132388B-C43 (D.G.O.), the Maria de Maeztu Units of Excellence Program CEX2020-001038-M (J.L.P.), the Aragon Government (E13-20R (D.S.) and E12-20R (J.L.C.)), the Programa Red Guipuzcoana de Ciencia, Tecnología e Innovación 2021 (Grant No. 2021-CIEN-000069-01, Gipuzkoa Next, J.I.P.), the Basque Department of Education (PRE-2021-2-0190 and PIBA-2020-1-0014, T.F.), and the Xunta de Galicia (Centro de Investigación accreditation 2019-2022, ED431G2019/03, D.P.)

## Author contributions

J.B. provided the concept. D.S., D.G.O. and J.I.P. supervised the project. D.S., J.B., N.M.D and A.B.L. performed the measurements and analyzed the data. J.L.C. assisted in data analysis. A.D.C. assisted in sample preparation. D.P. and M.V.V synthesized and characterized precursor molecules. S.S. and T.F. carried out the theoretical calculations. All authors discussed the results. D.S., D.G.O and J.I.P. wrote the manuscript.

## Competing interests

The authors declare no competing interests.
