## [Peer Review File · Nature Communications]

Reviewers' Comments:

Reviewer #1:

Remarks to the Author:

The manuscript of J. Brebe and co-workers reports on carbon-based magnetism, a hot topic in the last years. Indeed, the authors have several previous publications on the same topic. In this work, they present for very first time a direct measurement of the spin-polarization edge states in chiral graphene nanoribbons, demonstrating that electron correlations drive the spin-splitting of the electronic structure. So far, only indirect hints of the magnetism were obtained while the detection of the spin-polarization of pi-orbitals were elusive. The authors extract by means of spin-polarized scanning tunneling microscopy the energy-dependent spin-moment distribution from spatially extended edge states, which are stabilized by a ferromagnetic substrate. Therefore, this work is an important contribution to the field and should be published. However, there are several aspects in the manuscript that should be improved before acceptance:

1) Line 110: The authors claim that "The GNRs grow preferentially with their longitudinal axis along 110 high symmetry directions of the Gd atomic lattice, either the [110] (e.g. the central 111 ribbon in Fig. 1B) or the [211] directions of the Au(111) substrate." However, a random alignment can be observed in STM images. The alignment between the GNR and substrate is important in order to investigate the magnetic properties. Have the authors tried the growth approach of ref 35? In this work, on-surface synthesis of GNRs has been done directly on Au(111) followed by rare earth intercalation, yielding to GNRs on TbAu₂/Au(111). The ribbons and TbAu₂ are of high quality and aligned comparable with those directly grown on clean Au(111).

2) Can the size of the magnetic domains of the GdAu₂ be controlled?

3) Why is the in-plane magnetization of the substrate inverted when an external out-of-plane field is ramped from -3T to +3T? The authors write "we attribute the field dependent contrast to the appearance of a small in-plane projection of the external field (caused by the unavoidable slight misalignment of the surface normal with respect to the vertical axis of the STM head)." However, I do not think that the contribution of this small misalignment (<1 degree probably) would result in a clear contribution to the magnetic signal. I find the explanation of Supplementary Note 2 obscure and I think that this issue should be discussed with a couple of sentences in the main text.

4) For someone not familiar with SP-STM measurements it is not obvious why the magnetization of a Cr tip should not be affected by the external magnetic field. Cr tips are antiferromagnetic, therefore small changes in the tip apex can result in a change in the direction of the magnetization. A couple of sentences about the Cr tips and why have been used in the past to investigate magnetic systems should be included in the main text. In fact, it has been demonstrated that the magnetic moments at the Cr tip apex can be oriented along the vertical direction by an applied magnetic field of approximately 2 T (Japanese Journal of Applied Physics 51 (2012) 030208). In addition the spin polarization of a specific Cr tip is not a general property, as it can change significantly with a modification of the tip apex. Can the effect of the tip on the change in contrast in the SP-STM images be excluded after ramping the magnetic field? In addition, the Supplementary Note 1 reads: "The tips are submitted to voltage pulses until the expected in-plane spin contrast of 20 % or more at $V_b = 3$ V shown in Fig. S2B is obtained." As a result, the tip can be partially covered with GdAu₂ or can result in an unstable configuration. This issue should be discussed.

5) Are the images in Fig 2C and 2D taken with a CO-W tip or CO-Cr Tip. From the text (lines 145-147) reader can interpret that they were taken with a Cr-tip. However, in the caption of Fig 2 this is not clear. Is there any difference between the data taken with a CO-Cr tip and a CO-W-tip?

6) The CO functionalization of the Cr tips can be interesting for the community. The procedure should be explained in the Methods.

7) Do the data from Figure 3 correspond to two different GNRs? Or are they from the same GNR after switching the magnetic domain? How reproducible are these results?

8) Why the intensity distribution of edge states is not homogenous? The authors write: "This intensity distribution results from the combined effects of the LUMO confinement pattern at the edge and the influence of the moiré periodicity (Supplemental Fig. S4 and S5)." and in the Supplementary Note 1 "the edge state spatial distribution is heavily influenced by the moiré pattern of the underlying GdAu₂ monolayer. This is not related with the electronic structure of the edge states, but a consequence of the modulation of the local surface potential induced by the moiré, which gives rise to a periodically varying tunneling barrier." A difference in the local surface potential can result in a difference in the local doping of the ribbon. Is the charge transfer from the substrate not homogenous? This point is not clear.

9) Following the previous question. Is the inhomogeneity of the spin polarization of the edge a consequence of this? The author write: "On the other hand, the spin polarization of one edge is not homogeneous, contrary to the expectation for the ground state of isolated nanoribbons. This is ascribed to the hybrid character of the edge states, a consequence of electron doping and the orbital mixing associated to the U-term (Supplemental Fig. S3)."

Reviewer #2:

Remarks to the Author:

Manuscript of J. Brede et al. presents a direct observation of spin-polarized edge states in graphene nanoribbons. Spin detection with spatial resolution at the atomic scale is fundamentally important both for low-dimensional carbon-based magnetism and its applications. However, the detection of spin moments in these systems is quite challenging since the spin-orbit coupling is quite weak for light-element (like carbon) materials. In this work, authors demonstrate a unique approach to stabilize the remanent spin moment, allowing the detection using spin-polarized STM. The approach developed in this work for the detection of spin polarization might be extended to other carbon-based magnetic systems. Therefore, I would recommend the work by J. Brede to be published in Nature Communications after addressing the following points.

1. As shown in the topographic image in Fig. 1, graphene nanoribbons have different orientations with respect to the ferromagnetic substrate. While claimed by the authors, the magnetic coupling stabilizes the spin-polarization in graphene nanoribbon edges, it is not clear whether the spin-polarization depends on the orientation since the magnetic coupling might varies with the ribbon orientation.
2. Following previous points, authors claimed the magnetic coupling stabilizes the spin polarization in graphene ribbons but without direct experimental or theoretic evidence for such stabilization. Experimentally, for instance, comparing to the measurement in graphene ribbons on similar rare-earth metal surface alloys but without magnetic moments might verify the magnetic coupling role in the spin polarization. Alternatively, including the magnetic coupling into the calculation might allow to estimate the coupling strength in the system.
3. Authors claimed the observed spin contrast in the spin-polarized STS originates from the intrinsic spin polarization of graphene nanoribbon. It is not clear whether the hybridization with the metallic substrate contributes to the observed spin polarization. For instance, the calculated PDOS (Fig. 2E) does not match to the observed DOS (Fig. 2C). Discussion on such deviation might be desired to explain the influence from the substrate to the electronic structure and the spin textures of graphene nanoribbons.
4. In the calculated data (Fig. 3C), the spin asymmetry shows inverted contrast for edges of graphene ribbons. In the right edge, it shows positive value in the center and negative value for both ends, matching the measured spin contrast (Fig. 3B). In the left edge, the contrast is inverted in the calculated result. But in the measured data, it does not show similar spin contrast in two ends of the left edge (Fig. 3B). Authors should have some discussion on such disagreement with the calculation. Besides, Authors only present the spin asymmetry for energy below the Fermi level, despite some spin contrast is observed in the energy above the Fermi level. It should be more convinced with the comparison for various energies, at least the energies used in the measurement.
5. It seems the spin contrast are different for graphene nanoribbons if compared the results shown in Fig. 3 and Fig. S8. Does the spin polarization depend on the ribbon length?

Reviewer #3:

Remarks to the Author:

The field of carbon-based magnetism has made considerable progress since the bottom-up chemical self-assembly allowed synthesizing graphene fragments and nanoribbons with atomic precision. Yet, direct probing of spin-polarized states in graphene has remained an unsurmountable challenge due to the practically non-existent magnetic anisotropy. This work reports a scanning probe microscopy observation of the spin-polarized edge states in chiral graphene nanoribbons of finite length. This is a very thorough experimental work supported by simple, but well-connected calculations. It rests on two innovations: exchange coupling of the nanoribbons to the GdAu₂ ferromagnetic substrate and the combination of spin-polarized and CO-functionalized STM tips. I'm favorable to the publication of this manuscript in Nature Communications as the presented claim is sufficiently important and the research has been performed at a very high level.

That said, I believe one of the elements of this work is weaker than the others. This paper can be made much stronger if the authors show that the observed spin-polarization is not a result of the proximity effect with ferromagnetic substrate acting on otherwise (possibly) non-polarized edge states, and address the magnitude of the magnetic exchange between the spin-polarized edge states and the substrate. This could be done with the help of first-principles calculations. This remark should be taken as a recommendation, though.

Minor remarks:

1. The authors should include the theory counterpart of Fig. 2d.
2. Main text and the Methods section present two different justifications for reducing the magnitude of the Coulomb repulsion parameter U . Both make sense, but the two part of the text need to be brought in accordance.
3. The introductory part needs a proofread for English; it also uses ambiguous terminology. Please improve.

We thank the referees for the positive evaluation of our work and their interest in our results. Their perspective has definitely helped us to improve the article by adding new experimental data and getting a deeper insight on the relevance of our MFH calculations.

In the following we provide a point-by-point response to the reviewers' questions and comments, giving the exact positions of the revised main text where changes have been performed. We prefer this method to an annotated revised draft because, while the main claims of the revised version remain intact, the inclusion of additional discussions and figures has necessitated substantial editing and restructuring, making an annotated version of the changes in the manuscript rather inaccessible and hard to read.

We believe that we have been able to address all questions and requests and are confident that the work is now suitable for publication in Nature Communications.

Reviewer #1 (Remarks to the Author):

The manuscript of J. Brebe and co-workers reports on carbon-based magnetism, a hot topic in the last years. Indeed, the authors have several previous publication on the same topic. In this work, they present for very first time a direct measurement of the spin-polarization edge states in chiral graphene nanoribbons, demonstrating that electron correlations drive the spin-splitting of the electronic structure. So far, only indirect hints of the magnetism were obtained while the detection of the spin-polarization of pi-orbitals were elusive. The authors extract by means of spin-polarized scanning tunneling microscopy the energy-dependent spin-moment distribution from spatially extended edge states, which are stabilized by a ferromagnetic substrate. Therefore, this work is an important contribution to the field and should be published. However, there are several aspects in the manuscript that should be improved before acceptance:

1) Line 110: The authors claim that "The GNRs grow preferentially with their longitudinal axis along high symmetry directions of the Gd atomic lattice, either the [110] (e.g. the central 111 ribbon in Fig. 1B) or the [211] directions of the Au(111) substrate." However, a random alignment can be observed in STM images. The alignment between the GNR and substrate is important in order to investigate the magnetic properties.

Response: To analyse the ribbon alignment with respect to the substrate, we made use of images with simultaneous atomic resolution of the substrate and the ribbon's edge. We find alignments along four inequivalent crystallographic directions, which are:

-Chiral edge (i.e., ribbon axis) parallel to [-211]. Example revised Figure S6 (for edge atoms on top of Au row) and revised Figure S5 N=17 (edge atoms on top of a Gd row).

-Arm chair termini parallel to [-211]. Example revised Figure S5 N=12. Note that in this stacking, the ribbon's axis is 13° rotated with respect to the previous one.

-Chiral edge (ribbon axis) parallel to [1-10]_{Au}, that is, 30° tilted with respect to [-211]. Example Figure 1B.

-Armchair direction parallel to [1-10]_{Au}. We did not obtain atomically resolved images of this instance, but the [1-10]_{Au} direction is easily recognized in overviews with poorer resolution

because it is almost parallel to the moirée pseudo-lattice vectors. In our statistical analysis (see new Supplemental Figure S2), we do find GNRs at 13° from this direction.

If we consider these four directions, and taking into account the three-fold symmetry of the substrate, we can find ribbons along 12 different angles, which makes the orientation appear random at first sight.

Reviewer is right in remarking the relevance of the relative ribbon/substrate crystallographic arrangement for calculations. In this sense, our experiments suggest that chiral edge energetics prevail over the stacking of the internal carbon lattice. As shown in our general statistics (new Supp Fig. S2) the prevalence for sub-monolayer coverages is 51 % of ribbons with edges along high symmetry directions of the GdAu₂ (35 % and 16% along $[-211]$ and $[1-10]$ respectively) and 20 % of ribbons with internal graphene vectors parallel to high symmetry directions of GdAu₂ (i.e., the ones rotated 13° relative to the edge driven alignment).

New Supplementary Fig S2

To illustrate this preferential orientation of the GNRs' edge, we have added a new supplementary Figure S2 (see graph above) and reported its statistical analysis following the description of the on-surface synthesis procedure (lines 35-41 of the revised Supp. Info).

Have the authors tried the growth approach of ref 35? In this work, on-surface synthesis of GNRs has been done directly on Au(111) followed by rare earth intercalation, yielding to GNRs on TbAu₂/Au(111). The ribbons and TbAu₂ are of high quality and aligned comparable with those directly grown on clean Au(111).

Answer: During our initial calibration experiments we soon draw the conclusion that the quality of the (3,1,8)-chGNRs grown on GdAu₂ was similar (if not better) than those grown on Au(111) under the same parameters. Therefore, it was not justified the use of the intercalation technique. See Figure R1 for a comparison of the cGNR quality in both substrates.

Figure R1.- Comparison of the synthesis of cGNR on Au(111) and GdAu₂/Au(111), $V_b=1V$. The only difference in the procedure is the need of a slightly higher optimum cyclo-dehydrogenation temperature on GdAu₂ than on bare Au surface (340 °C and 320 °C respectively). A fake colour scale mixing the topography with its Laplaced transform has been used in order to enhance the visibility of the GNRs boundaries.

2) Can the size of the magnetic domains of the GdAu₂ be controlled?

Answer: There are two types of walls delimiting magnetic domains. We can find the *natural* domain walls typical of a soft in-plane ferromagnet, characterized by a gradual rotation of the magnetization from one domain to the other across a characteristic length scale, the domain wall width. The domain wall width in GdAu₂ from our SP-STM measurements varies between 5-10 nm, as can be seen in Figures S2B, S9A and S13A of the revised supplemental information. There are also sharp domain walls occurring at the structural antiphase boundaries (APBs), which are discussed in the context of Supplementary Fig. S3. These boundaries appear when two structural domains shifted by half a unit cell of the Gd lattice meet each other. As shown by spin resolved measurements in Fig. S3, these APBs are featured by a local antiferromagnetic coupling across the domains, with a sharp transition between opposite magnetization directions which is just one atomic row wide. These can be considered artificial domain walls.

The formation of natural domains walls depends on the density of APBs, because the magnetostatic energy gain associated to natural domain walls is controlled by the constrain of antiferromagnetic coupling across the APBs. The density of APBs happens to be controlled by the substrate temperature during Gd deposition. At a temperature of 350 °C there are nearly no APBs, and natural domain walls are relatively large and very scarcely found. Reducing the temperature to 320 °C increases their density to an optimum value for our purposes, with long APBs and natural domain walls of about 10 nm (Fig S9A). As discussed in lines 117 to 121 of the revised article, we need enough density of APBs as to get well defined magnetic contrast in relatively small regions next to the target GNRs. This permits to constantly monitor the tip and surface magnetization (see supplementary. Fig. S9) without compromising the tip stability (no need of navigating large distances to characterize the tip sensitivity direction and confirm the reversal of the substrates magnetization underneath the ribbon).

Consequently, the size of the magnetic domains can be controlled by the substrate temperature during deposition. They are enclosed by artificial APBs domain walls, whose density depends on that temperature, and by natural domain walls, whose width also depends on the local distribution of APBs.

3) Why is the in-plane magnetization of the substrate inverted when an external out-of-plane field is ramped from -3T to +3T? The authors write “we attribute the field dependent contrast to the appearance of a small in-plane projection of the external field (caused by the unavoidable slight misalignment of the surface normal with respect to the vertical axis of the STM head).” However, I do not think that the contribution of this small misalignment (<1 degree probably) would result in a clear contribution to the magnetic signal. I find the explanation of Supplementary Note 2 obscure and I think that this issue should be discussed with a couple of sentences in the main text.

Answer: $3 \text{ Tesla} * \sin(1^\circ) = 53 \text{ mT}$. The in-plane coercive field characterized by SP-STM with in-plane magnetic fields is of the order of 20 mT [1]. Thus, we apply a significant in-plane field, which is responsible for the abrupt switching due to the displacement of natural domain walls in a soft ferromagnet.

As further discussed in the answer to question #4 below, the out-of plane projection is able to overcome the magnetic anisotropy of the system and align the magnetization out of plane (slow saturation at large fields in Fig S8C), but the measurements are performed in remanence, with the substrate magnetization in-plane. Its sign can be reverted by smaller fields as the domains with opposite orientations grow for different signs of the in-plane field (which just follows the sign of the out-of-plane main component).

Following the reviewer’s advice, we have clarified this issue along lines 121 to 128 of the revised article. We believe that including additional details about the tip-sample magnetic state at elevated fields is not appropriate for the main text, since all our results are obtained at remanence. Therefore, we only report in the main text the way we achieve the different remanent states, and leave the continuous characterization of the tip magnetic state for the supplementary note 2 and revised Fig. S8.

4) For someone not familiar with SP-STM measurements it is not obvious why the magnetization of a Cr tip should not be affected by the external magnetic field. Cr tips are antiferromagnetic, therefore small changes in the tip apex can result in a change in the direction of the magnetization. A couple of sentences about the Cr tips and why have been used in the past to investigate magnetic systems should be included in the main text. In fact, it has been demonstrated that the magnetic moments at the Cr tip apex can be oriented along the vertical direction by an applied magnetic field of approximately 2 T (Japanese Journal of Applied Physics 51 (2012) 030208). In addition the spin polarization of a specific Cr tip is not a general property, as it can change significantly with a modification of the tip apex. Can the effect of the tip on the change in contrast in the SP-STM images be excluded after ramping the magnetic field? In addition, the Supplementary Note 1 reads: “The tips are submitted to voltage pulses until the expected in-plane spin contrast of 20 % or more at $V_b = 3 \text{ V}$ shown in Fig. S2B is obtained.” As a result, the tip can be partially covered with GdAu₂ or can result in an unstable configuration. This issue should be discussed.

We agree with the reviewer that the spin-polarization of Cr tip cannot be viewed as a general property. We select our working tips by their function, rather than by their design, and that is the reason why we keep track of the tip performance during our experiments (see for example revised Fig. S9, where the magnetic state of the tip is constantly monitored during the experiment discussed in Fig3, to conclude that the tip sensitivity remains constant).

Cr-tips can show various saturation fields and sensitivity directions. The reference cited by the reviewer [2] demonstrates a saturation field of about 3 Tesla and in-plane sensitivity direction at zero field. There are also examples with a different behaviour, like in reference [3], where out-of-plane sensitive Cr-tips with no response to fields as large as 6 Tesla are used. Their main feature at the end of the day is that one can prepare tips relatively insensitive to the external magnetic field in the range of several Tesla, which is the most convenient choice to characterize soft ferromagnets such as GdAu₂. To make this point clear for readers not familiar with SP-STM, we have included a brief justification of our choice for bulk Cr-tips along lines 114 to 116 of the revised article.

We also specify at the Methods section (lines 512 to 517) the conditions that have to be met for a particular tip in our experiments. Our procedure (electrochemical etching 2M NaOH followed by in-situ field emission at high voltage) ends up with tips that not always exhibit SP-STM contrast across the APB. We submit the tip to additional voltage pulses (positive bias between tip and sample) until this contrast appears. The spin-polarization and quantization axis of the Cr tip apex will depend on atomic scale details of its structural arrangement in the bulk Cr rod from where it is suspended. Voltage pulses are able to alter this structure until a quantization axis with significant in-plane component is obtained, for which SP-STM contrast across APBs is then readily obtained. As long as deliberate indentation of the tip into the GdAu₂ is avoided, the slow saturation vs. field of the magnetic contrast over GdAu₂ shown in Fig. S9 is indeed a general characteristic. This gradual saturation can be ascribed to a coherent spin rotation of the GdAu₂ magnetization towards its hard magnetization axis, or to a canted easy magnetization axis of the tip apex at zero field, which aligns gradually with the external field above 2-3 Tesla. Although this dilemma does not have an impact in our main analysis in remanence, the symmetric behaviour of the spin contrast of neighbouring domains (varying with the trend of approaching their dI/dV signals to the average value of their maximum spin contrast at zero field) indicates that gradual variation should be ascribed to the rotation of the GdAu₂ spins and not to the tip. This is in agreement with XMCD data of the Gd M₅ adsorption edge, which shows hard axis saturation above 2 Tesla for bare GdAu₂ [4].

We cannot discard that some of our tips are not pure Cr and may contain some atomic clusters of GdAu₂. We have, however, good reason to believe that this is not the case for freshly etched and field emitted tips. If the as-prepared tips are gently dipped into the surface several times, their spin polarization starts showing sign reversals at low fields (simultaneous contrast reversal of regions that are not magnetically coupled, like the example in the new Supp. Fig. S2B), and the saturation pattern discussed in Fig. S9 changes. Tips in this unstable condition are dismissed for the characterisation of the GNRs.

Finally, we would like to comment that we put an extra effort into using tips with consistent magnetic behaviour as a function of field, as well as of their spin resolved contrast in spectroscopy.

5) Are the images in Fig 2C and 2D taken with a CO-W tip or CO-Cr Tip. From the text (lines 145-147) reader can interpret that they were taken with a Cr-tip. However, in the caption of Fig 2 this is not clear.

Answer: Lines 145-147 of the previous version referred to the fact that we do not know what kind of spin-contrast we can expect from a CO-functionalized Cr-tip, and thus we have to be able to access the edge states with a metallic tip (Cr or W). We thank the reviewer for bringing about this comment because the kind of tips used for Fig 2 were certainly not clear. We have

rewritten the lines mentioned by the reviewer stating clearly the tip nature (lines 143 and 151).

The whole data set of figure 2 was taken with a bare W tip, except the high resolution current panel Fig. 2A, for which we specify that the W tip was functionalized with CO. We expect that the new text and the caption format now is clear for the readership.

Also we have labelled all images (in the main article and supplemental information) where the tip termination is of interest as Cr, CO-Cr, W or CO-W, as appropriate.

Is there any difference between the data taken with a CO-Cr tip and a CO-W-tip?

There is not qualitative difference, see Figs S6 (W-CO tip) and S8 (Cr-CO tips). Since the CO tips are less stable than the bare Cr tips, we did not complete any full magnetic cycle with CO-Cr and do not have information about their ability to probe magnetic contrast.

6) The CO functionalization of the Cr tips can be interesting for the community. The procedure should be explained in the Methods.

Answer: The functionalization of the Cr tip proceeds rather similar to the functionalization of W tips covered with coinage metals or bare W tips. In this technique, the main differences are introduced by the substrate itself. Below we attach Figure R2 for review purposes (which could be added into the supplemental information if requested). The first notable difference with the case of CO adsorbed on coinage metals is that it appears as protrusion (Fig R2c) instead of the characteristic sombrero. Inelastic vibrational excitations of the CO adsorbed on GdAu₂ are found at 2 meV and 23 meV (Fig R2a), which are a factor 2/3 smaller than the excitations thresholds on Au and Cu surfaces. For the particular example of Fig R2b-c, when the CO is transferred to the Cr-tip, the excitation thresholds increase to 3 meV and 33 meV respectively.

The functionalization protocol is the same as on Au₁₁₁ or NaCl, easier and with larger success rate than in the former, and more difficult and with lower success rate than in the latter. As shown in Fig. R2c, the CO jumps to the Cr-tip with the following procedure, which is now reported in the revised Methods section (lines 518 to 521): first a sharp metallic tip needs to be prepared, then we scan one line at low gap resistance (2-20 mV and 2-5 nA) in constant current mode until an abrupt jump in topography is detected, and then restore normal scanning conditions. If the CO has been transferred to the tip, we get an immediate improvement of the resolution (middle panel Fig. R2c) and the CO is no longer visible (right panel).

Figure R2. Cr-tip functionalization with CO. a,b) Inelastic signal of CO in panel c adsorbed on GdAu₂ surface (a) and the same CO adsorbed on the Cr tip over the GdAu₂ surface (set point: 20 mV, 0.5 nA; modulation 1 mV rms). c) Images of the CO/GdAu₂ system before and after tip functionalization by increasing the tunnelling set-point to 5 nA when the scanner passes over the adsorbate (regulation 20 mV, 0.5 nA).

7) Do the data from Figure 3 correspond to two different GNRs? Or are they from the same GNR after switching the magnetic domain? How reproducible are these results?

Answer: They are the same GNR at two different energies (8 meV below Fermi level and 5 meV above Fermi level). The sign change of the spin polarization across Fermi level, which follows nicely the energy resolved spin polarization shown in panel E of the same figure, it is one of the main characteristics of an e-e correlations driven spin splitting. This is clear in regions 1, 2 and 3, which are the segments of the ribbon where intense edge-state with double peak structure around the Fermi level are found (see our reply to comments #8 and #9 of reviewer #1).

Regarding the reproducibility, these data are fully reproducible. To establish the method and the magnetic phenomenology in the original context of GNR edge states, we only considered data in which the magnetic contrast of the ribbon can be systematically reversed several times. This implies changing between the AP and P configurations represented in Fig. 1G at least twice, for instance AP→P→AP. Then we plot the spatial distribution of the spin polarization, by computing the $(AP-P)/(AP+P)$, where AP and P are the dI/dV of the magnetic states with opposite substrate magnetization. In the case of the data in Fig. 3, we did an additional cycle (revised supplemental Fig S9). In this way, the magnetic contribution of the signal becomes

unambiguously determined by subtracting any P from any AP state, and the ultimate reproducibility and tip stability is demonstrated by checking that there is no variation between any two “P” states or “AP” states (see revised Supplementary Fig S10, formerly S7). Same systematics were followed in the additional examples shown in revised supplementary Fig. S13 and new Fig. S14.

To make this procedure clearer, we have added the calculation formula using the notation described in the main text in the colour scale of Fig 3B, and we have put the theoretical calculation of the magnetic moment Fig. 3C in exactly the same colour scale.

8) Why the intensity distribution of edge states is not homogenous? The authors write: “This intensity distribution results from the combined effects of the LUMO confinement pattern at the edge and the influence of the moiré periodicity (Supplemental Fig. S4 and S5).” and in the Supplementary Note 1 “the edge state spatial distribution is heavily influenced by the moiré pattern of the underlying GdAu₂ monolayer. This is not related with the electronic structure of the edge states, but a consequence of the modulation of the local surface potential induced by the moiré, which gives rise to a periodically varying tunneling barrier.” A difference in the local surface potential can result in a difference in the local doping of the ribbon. Is the charge transfer from the substrate not homogenous? This point is not clear.

Following the referee's advice, we make this point clearer. This apparent controversy is associated to the complex interplay of the moiré effect and the edge state. Therefore, we have rewritten in a more explicit manner the discussion of this issue in the main text along lines 139-147 of the revised version, and removed the sentence quoted by the reviewer in lines 227-229 of the previous version (in this way, the experimental spin resolved data is first presented and, subsequently, the deviation from expectation for the isolated GNR is discussed in the context of our electron doping model).

We have also revised Supplementary Note 1 as follows:

Typical dI/dV images of GNRs are obtained in constant height conditions, because it provides a more quantitative picture of the LDoS than constant current imaging, which is affected by the crosstalk between the feedback regulation and the dI/dV intensity. Using the same line of reasoning as in the main text, we first remark that metallic tips are not able to detect the edge state in constant height conditions for tip-ribbon distances above the manipulation threshold (i.e., at a distance in which the ribbon is not displaced laterally by the forces exerted by the tip). In contrast, CO-functionalized tips can probe this edge intensity [5], because they are less reactive and allow for smaller tip-sample distances without perturbing the target ribbon (Fig. 2A, revised Fig. S6A and revised Fig. S8). However, constant height (CH) conditions are sensitive to the variations of the local tunnelling gap, which at first approach (simple plane waves model) is determined by the tip-surface distance (z) and by the and the local surface potential (ϕ_s). As a consequence, the periodic variations of the tunnelling gap ascribed to the moiré, are mirrored in the distribution of the edge state intensity retrieved by the CO tip in CH conditions. In summary, in CH conditions, metal tips do not detect the edge state, whereas CO tips can do it, but the longitudinal distribution of the intensity is dominated by the moiré, which hinders the true structure of troughs and ridges expected for the quantum well states confined within the length of the GNR. In this sense, the edge contrast retrieved by CO tips in CH conditions, is not proportional to the LDoS along the edge (see revised Fig. S6C).

To demonstrate the ϕ_s variation ascribed to the GdAu2 moiré, we performed measurements of its field emission resonances (FER), which are presented in new Supplemental Figure S5 showing the correlation of ϕ_s with the moiré structure. A modulation of around 50 meV can be clearly observed in the 3rd to 6th FERs (panel C) between the bright parts and the dark triangles of the moiré. This shift is an approximation of the variation of ϕ_s . A better estimation can be retrieved from the back extrapolation to zero of the FER bias vs. the FER order to the power of 2/3 [6], Fig. S5D, which gives $\phi_s=4.71\pm 0.14$ eV and $\phi_s=4.55\pm 0.15$ eV for brightest and darkest topographic moiré spots respectively. The full 2D correlation can be better appreciated in dI/dV maps tuning the bias to the exact energy of the 2nd FER at position of the darkest part of the topography (the centre of any triangle in panel A), $V_b=6.65$ V. In this way, panel B shows an inverted map of ϕ_s where the brightest regions correspond to a FER at exactly this energy, and the darkest regions are the same FER slightly shifted upwards in energy (and thus with higher ϕ_s), see dI/dV spectra in panel C.

If the surface topography z or ϕ_s fluctuate in the image frame, the constant height current intensity is going to pick the same fluctuation. The periodically varying tunnel barrier translates directly into the same pattern in constant height images of the GNR, for both metallic and CO tips (see Figures 1H, 2A, 3A, and new S6A, S8 and S13B, a collection of combined data from metallic and CO functionalized tips), and this should not be regarded as an intrinsic modulation of the edge state.

Nevertheless, we believe that the inhomogeneous ϕ_s indeed causes in the GNR a deviation from the charge distribution expected from the pure quantum well states of the free standing case. The isolated GNR and its edge states are well described by the MFH Hamiltonian. In this picture, the moiré fluctuations would behave as a perturbation of this Hamiltonian. As such, the perturbed eigenstates have slightly different energies and LDoS distribution. For instance, the perturbation associated to the ϕ_s variation will increase the population in parts of the edge states (see new Supp. Fig. S7) over maxima of the surface potential, and deplete parts of the edge states over minima. This will have an impact in the local charge distribution specially around Fermi level. So we coincide with the referee that charge transfer to the edge state is not homogeneous in the sense that the interaction with the substrate modifies the charge distribution of the free standing ribbon, although the doping does not occur at a local level (implying by 'local' a dimension smaller than the ribbon length).

New Supplemental Figure S5

9) Following the previous question. Is the inhomogeneity of the spin polarization of the edge a consequence of this? The author write: “On the other hand, the spin polarization of one edge is not homogeneous, contrary to the expectation for the ground state of isolated nanoribbons. This is ascribed to the hybrid character of the edge states, a consequence of electron doping and the orbital mixing associated to the U -term (Supplemental Fig. S3).”

Response: We agree with the reviewer that the correlation between the inhomogeneous surface local potential and the distribution of the spin polarization deserves further discussion. We believe that there is an indirect correlation, but our MFH calculations indicate that this is not the main underlying mechanism for the inhomogeneous spin polarization.

The first allowed confined edge state of the conduction band in a charge neutral 3,1,8-cGNR is expected to show an intensity pattern like a broad cusp and no internal nodal points along the edge. This is a rather homogenous charge distribution. If this state is subject to e-e correlations of the appropriate strength, it will display as well a rather homogenous spin density along the edge (see new Supplemental Figure S12A). Increasing the on-site Coulomb Repulsion U in our MFH Hamiltonian (eq. 1 in Theoretical Methods) mimics the effect of enhanced e-e correlations in the real system. The charge neutral case has particle-hole symmetry, and the gradual increase of U will result in a concomitant increase of the correlations splitting, increasing the experimental spin polarization as the splitting exceeds the FWHM of the molecular orbitals.

When the GNR becomes electron doped by the transfer from the substrate, the particle-hole symmetry breaks down. Now the effect of increasing U (increasing e-e interactions) is not easy to predict. Revised supplementary Fig. S4 (formerly S3) shows, for the case of 2 electrons added, how the three first confined edge states of the uncorrelated GNR ($U=0$) mix up for increasing U . The new eigenstates of the system are now hybrids of the non-interacting confined states. This effect is triggered by the e-e correlations and caused by the electron doping as a necessary condition. Since the mixed eigenstates have a different structure of crests, troughs and nodes, it is expected that the spin polarization will also display oscillations ascribed to this mixing. This is clearly shown in our simulations of the spin polarization at $U=1.0$ eV for one and two electrons transferred from the substrate, Figures S12B,C. The spin polarization shows one and two changes of sign along the edge for one and two added electrons, respectively.

New Supplemental Figure S12

As discussed above, the modulation of the surface potential alters the charge distribution of the edge states, and can have an impact in the strength of e-e correlations and its associated splitting that allow (or not) for the spin-polarization of the GNR states and determine the finally observed spin-contrast. Nevertheless, although we believe that this modulated surface potential may indeed induce an inhomogeneous distribution of the transferred charge, this does not seem to be the most relevant parameter determining the spin polarization. Actually, the oscillatory spin polarization along each of the edges is captured by our calculations of GNRs in the gas phase, without additional considerations beyond the total amount of charge transfer.

The modulation of the surface potential can however induce depletion or accumulation of charge, acting as an additional perturbation of the MFH model (see answer to comment #8). For instance, the spin polarization at the upper half of the left edge in Figure 3B clearly departs from the calculated pattern in Fig. 3C. At the same time, the spin averaged LDoS (Fig 3D) in this region lacks the intensity at Fermi level found at the bottom half. In all our spin-resolved experiments, there is a one to one correspondence between the appearance of a split-peak at Fermi level, and the finite spin polarization detected at the energies of the sub-peaks. The spin polarization emerges in line with the MFH calculations of free standing GNRs whenever there is a spin-split peak saddling the Fermi level. This suggests that the effect of the surface potential modulation is a charge redistribution with respect to the free standing GNR, but does not play a significant role in the emergence of the magnetization. Note that there is no a clear correlation between the moiré pattern in Figs. 2A and 3A, and the appearance of a large split intensity at Fermi level (Figs 2D and 3D). Thus, understanding the influence of the inhomogeneous surface potential in the charge redistribution requires a detailed (and computationally expensive) theoretical description of the surface and subsurface layers, which, since it is not the main ingredient for the spin polarization, we think that it is beyond the scope of this work.

To make this scenario more accessible to the reader, we have grouped in the same paragraph (lines 276 to 285 of the revised version) the discussion of the hybridization caused by the combined effect of electron doping and $U \neq 0$ (formerly lines 180-183), with its impact over the spatial arrangement of the spin polarization (formerly lines 261-266).

Reviewer #2 (Remarks to the Author):

Manuscript of J. Brede et al. presents a direct observation of spin-polarized edge states in graphene nanoribbons. Spin detection with spatial resolution at the atomic scale is fundamentally important both for low-dimensional carbon-based magnetism and its applications. However, the detection of spin moments in these systems is quite challenging since the spin-orbit coupling is quite weak for light-element (like carbon) materials. In this work, authors demonstrate a unique approach to stabilize the remanent spin moment, allowing the detection using spin-polarized STM. The approach developed in this work for the detection of spin polarization might be extended to other carbon-based magnetic systems. Therefore, I would recommend the work by J. Brede to be published in Nature Communications after addressing the following points.

1. As shown in the topographic image in Fig. 1, graphene nanoribbons have different orientations with respect to the ferromagnetic substrate. While claimed by the authors, the

magnetic coupling stabilizes the spin-polarization in graphene nanoribbon edges, it is not clear whether the spin-polarization depends on the orientation since the magnetic coupling might vary with the ribbon orientation.

Answer: With regard to the GNR orientation, we refer to the reply to reviewer #1.1. Among these preferential orientations, we have obtained unambiguous magnetic contrast in 5 different GNRs, oriented along [-110] (for example Fig 3), 13° tilted with respect to [-110] (not shown), and along [-2,1,1] (new Supp Fig. S14 for example). We do not have enough statistical information to establish a correlation between the orientation and the magnetic coupling, other than concluding that the orientation is not decisive in obtaining an effective magnetic coupling. The exact stacking of carbon atoms on top of the Gd lattice should also play a role. This is not always the same even for identical orientations, for instance, revised Fig. S6 shows a GNR with the edge on top of a Au row, while Fig. S8 has the edge on top of a Gd row.

Then, the exchange coupling between Gd spin and GNR spin is probably a non-trivial function of the superposition of the two lattices. However, the role of this interaction is to stabilize the magnetic moment of the GNR edge, not to create it. The oscillatory spin polarization reported in Fig. 3 and in the supplementary Fig S14 is characteristic of the ribbon (we refer to our answer to comment #9 of reviewer #1), and under the same conditions of screening and electron doping, it will be the same regardless of the magnetic state of the substrate (or even for non-magnetic substrates).

Therefore, there is not a correlation of the spin polarization with the site dependent magnetic coupling. To be precise, the like a magnetic coupling and the orientation determine the *capability* to probe the spin polarization of the GNR. If the overall energy of the system does not depend on the relative magnetization directions of the GNR and the GdAu₂, the spin moment of the GNR would be fluctuating paramagnetic atom over time, and the slow SP-STM technique would sense all equally probable magnetization directions, yielding a mean value of zero. We have measured two additional GNRs following the same procedure without spin polarization. One possible explanation is that some particular stacking patterns of the GNR and Gd lattices provide a very small exchange energy (< thermal fluctuations of 1.2 K). Strictly speaking, without the stabilizing GdAu₂, the GNR would only have a stationary spin-polarization at absolute zero temperature and in the absence of quantum spin fluctuations. In this sense, the effect of the GdAu₂ can be seen as an overall mean field exchange interaction which creates a preferential stable direction for the intrinsic magnetization of the GNR.

We have rewritten the discussion related to the stabilization of the chGNR spin polarization, making clear the key technical role of the magnetic interaction with the substrate in regard to the definition of the magnetic quantization axis and a finite stationary value of S_z (lines 302 to 312).

2. Following previous points, authors claimed the magnetic coupling stabilizes the spin polarization in graphene ribbons but without direct experimental or theoretic evidence for such stabilization. Experimentally, for instance, comparing to the measurement in graphene ribbons on similar rare-earth metal surface alloys but without magnetic moments might verify the magnetic coupling role in the spin polarization. Alternatively, including the magnetic coupling into the calculation might allow to estimate the coupling strength in the system.

The reviewer probably means the alloy YbAu₂, where Yb has a full 4f shell and therefore no magnetic moment. Due to the negligible magneto-crystalline anisotropy of carbon materials,

the chGNRs have no reason to exhibit a stationary finite magnetization over a non-magnetic or paramagnetic material. Under these conditions, spin-polarized STM (or other techniques with poor time resolution like XMCD) would not be able to probe the chGMR magnetization.

Instead, we note that the coupling is demonstrated by the studies of spin polarization in chGNRs crossing natural domain walls of the GdAu2 (revised supplementary Fig. S13, formerly Fig. S8). Here, one end of the GNR acts as a reference spin polarization, which allows to probe the opposite end and confirm that the magnetization of each end is driven by the spin direction of the GdAu2 region underneath. This is a complex experiment which was not well explained in the previous version. Consequently, we have included an arrow sketch (similar to the one in Fig. 1) in Supplementary Fig. S13.

Notwithstanding the above, we can give a bottom threshold for the interaction strength. While all presented data up to now are taken at 1.2 K, we were also able to measure a reliable spin polarization of the same approximate value (~5 %) at a temperature of 4.3 K. This data is shown in new Supp. Fig. S14 for a chGMR with a length of 26 precursor units. This means that exchange interaction between the chGMR and the GdAu2 is at least larger than the thermal energy corresponding to 4.3 K, which amounts to 0.4 meV.

New Figure S14

3. Authors claimed the observed spin contrast in the spin-polarized STS originates from the intrinsic spin polarization of graphene nanoribbon. It is not clear whether the hybridization with the metallic substrate contributes to the observed spin polarization. For instance, the calculated PDOS (Fig. 2E) does not match to the observed DOS (Fig. 2C). Discussion on such deviation might be desired to explain the influence from the substrate to the electronic structure and the spin textures of graphene nanoribbons.

Answer: The reviewer is right in highlighting this important issue, because the proximity of the GNR to the ferromagnetic surface could also induce some spin-polarization. There are, however, three critical empirical evidences against it, which point to a spin polarization that is intrinsic to the GNR (and are addressed along lines 237 to 291): 1) the spin polarization of GdAu2 is *energy independent* (Supplementary Fig. S2) and very small (<4 %) in the energy range relevant for the GNR, in sharp contrast with the sign reversal of the spin polarization of the GNR edge (with peaks of ± 8 %) crossing the Fermi level in a range of only 10 meV. 2) the spin polarization is circumscribed exclusively to the edge, while a proximity effect should give rise to spin polarization in the entire GNR structure. 3) The oscillatory pattern of the spin polarization is qualitatively the one expected for a GNR charged with 2 additional electrons (Fig. 3 and new Supplementary Fig. S11).

Since there have been several other questions in this line, we refer to the answer to question #9 of reviewer #1. There, it is discussed that the modulation of the local surface potential can act as a perturbation for the magnetic ground state of the chGNR, but it does not quench the main intrinsic characteristics of the chGNR magnetism listed above. This is now explicitly reported along lines 292 to 301 of the revised article.

We believe that the calculated PDOS (Fig 2E) for 2 e- added and $U=1$ eV matches quite reasonably the measured one, especially for the three most relevant states for the magnetism (HOMO, LUMO and LUMO+2 in the notation of the charge neutral specimen). As stated along lines 163-193, our calculation nails down the physical description of the experiment: A peak that belongs to the conduction band of the charge neutral chGNR appears at Fermi level, surrounded by the first quantum well state of the valence band and the next order quantum well state of the conduction band. The peak at Fermi level is almost at the onset of the CB and splits in two for finite on-site Coulomb repulsion. In good agreement, the Fourier analysis discussed in Fig. S6 and the real space nodal distribution of simulated molecular states (new Fig. S7), clearly show that the onset of the CB is below the Fermi level and the VB is located around -40 meV.

4. In the calculated data (Fig. 3C), the spin asymmetry shows inverted contrast for edges of graphene ribbons. In the right edge, it shows positive value in the center and negative value for both ends, matching the measured spin contrast (Fig. 3B). In the left edge, the contrast is inverted in the calculated result. But in the measured data, it does not show similar spin contrast in two ends of the left edge (Fig. 3B). Authors should have some discussion on such disagreement with the calculation.

Answer: The calculation method does not allow fractional charge doping for the case of finite nanoribbons. In the case of even number of additional electrons, the charge is distributed evenly among the two spin channels (see methods section), which in combination with the inversion symmetry of the structure, leads to a total magnetic moment of zero ($S_z=0$). This

does not happen in the real GNR hybridized with the GdAu2 surface, as is now explicitly mentioned at the theoretical methods section and along lines 285-291. The total energy if the two electrons are added in the same spin subband (and thus $S_z=1$) is 3 meV higher.

As discussed above for remark #3, and also in the context of the answer to question #8 of reviewer #1, the effect of the ~ 50 meV fluctuation of the local surface potential is not included in our calculations. We are positive that this fluctuation can give rise to a redistribution of the charge density along the edge, modifying but not fully adulterating the distribution of the ideal quantum well state (because of the qualitative agreement of the experiments with a relatively simple MFH model). If a given state suffers a DoS reduction at a given position, the correlations splitting will not be there either, and will also suppress the spin polarization.

Furthermore, there is the possibility that the strength of the exchange interaction of all chGNR sites with magnetic moment is not the same. As shown in Fig. 3B, new Supplementary Fig. S14 and other additional data sets (not shown), the edge states display a minimum spin correlation length of about 6 precursor units. If the spin correlation length is smaller than the length of the GNR, we can meet a situation in which a part of the edge is stabilized by the proper exchange interaction with the GdAu2, while the opposite end is not.

We do not have enough statistics as to ascertain which one of the two possible explanations given above are responsible for the deviation with respect to the MFH calculation. In the revised article we treat the two possibilities as “extrinsic factors” along lines 292-301.

We would like to stress that the purpose of this work is to establish the existence of a correlations driven spin polarization in the edge states, and report the technique to study similar physics in other graphene nanostructures. We believe that the in depth study of the chGNR interaction with REAu2 alloys is a different subject.

Besides, Authors only present the spin asymmetry for energy below the Fermi level, despite some spin contrast is observed in the energy above the Fermi level. It should be more convinced with the comparison for various energies, at least the energies used in the measurement.

Answer: We now include the calculated spin-asymmetry for unoccupied states in theoretical panel of the Fig. 3C. It can be seen that, as corresponds to a e-e correlations driven splitting, the unoccupied peak shows a different third component of the spin moment density. In agreement with the experiment in some regions (regions 1 and 3 in Figs. 3B and 3C), the spin polarization is reversed when the Fermi level is crossed.

Note that in order distinguish the different orbitals laying very close in energy (as is the case of the occupied and unoccupied states in Fig. 3C), we have decreased the value of γ (see equations 2-4 in Methods section) from 30 to 1 meV.

To show the variations of the spin distribution with energy in the case of free standing nanoribbons with integer charge transfer, we have included new supplementary Figure S11.

5. It seems the spin contrast are different for graphene nanoribbons if compared the results shown in Fig. 3 and Fig. S8. Does the spin polarization depend on the ribbon length?

Answer: Longer ribbons will have smaller energy spacing and thus be prone to accept more electrons than shorter ones. This will affect the mixing of quantum well states and thus the

spin polarization pattern (see for instance new supplementary Fig. S12). However, for the lengths addressed here (12-20 precursor units), all GNRs fall in line with a doping level approximately two electrons, and for this case all ribbons display the same kind of oscillatory spin polarization with two sign changes (for sufficiently large U). The case of the revised Fig. S13 (formerly Fig. S8) is different. Here, we positioned the GNR across a natural domain wall. The substrate's spin polarization there is not homogeneous at all, varying from spin up to spin down with all intermediate possible orientations. This leads to a large degree of frustration of the chGNR magnetic moment density, as opposed to the collinear model applicable to Fig. 3. As discussed in the answer to question #2, the purpose of the study with inhomogeneous GdAu2 magnetization is not to back the conclusions from Fig. 3, but to prove the existence of sizable exchange coupling between the GNR and the substrate. This is a complex experiment which was not well explained in the previous version. Consequently, we have included an arrow sketch (similar to the one in Fig. 1) in Supplementary Fig. S13.

Reviewer #3 (Remarks to the Author):

The field of carbon-based magnetism has made considerable progress since the bottom-up chemical self-assembly allowed synthesizing graphene fragments and nanoribbons with atomic precision. Yet, direct probing of spin-polarized states in graphene has remained an unsurmountable challenge due to the practically non-existent magnetic anisotropy. This work reports a scanning probe microscopy observation of the spin-polarized edge states in chiral graphene nanoribbons of finite length. This is a very thorough experimental work supported by simple, but well-connected calculations. It rests on two innovations: exchange coupling of the nanoribbons to the GdAu2 ferromagnetic substrate and the combination of spin-polarized and CO-functionalized STM tips. I'm favorable to the publication of this manuscript in Nature Communications as the presented claim is sufficiently important and the research has been performed at a very high level.

That said, I believe one of the elements of this work is weaker than the others. This paper can be made much stronger if the authors show that the observed spin-polarization is not a result of the proximity effect with ferromagnetic substrate acting on otherwise (possibly) non-polarized edge states,

We agree that first-principle calculations would shed light in many interesting details of the GdAu2-GNR magnetic interaction. However, as discussed in other parts of the response, we think that the cornerstone of the work is the spin polarization reversal crossing the Fermi level, which cannot realistically be attributed to an interaction with the ferromagnetic substrate. In this sense, we copy paste below the answer to question #3 of reviewer #2, verbatim,

'There are, however, three critical empirical evidences against it, which point to a spin polarization that is intrinsic to the GNR (and are addressed along lines 237 to 291): 1) the spin polarization of GdAu2 is *energy independent* (Supplementary Fig. S2) and very small (<4 %) in the energy range relevant for the GNR, in sharp contrast with the sign reversal of the spin polarization of the GNR edge (with peaks of ± 8 %) crossing the Fermi level in a range of only 10 meV. 2) the spin polarization is circumscribed exclusively to the edge, while a proximity effect should give rise to spin polarization in the entire GNR structure. 3) The oscillatory pattern of the spin polarization is qualitatively the one expected for a GNR charged with 2 additional electrons (Fig. 3 and new Supplementary Fig. S11).'

And address the magnitude of the magnetic exchange between the spin-polarized edge states and the substrate. This could be done with the help of first-principles calculations. This remark should be taken as a recommendation, though.

We note that the coupling is demonstrated by the studies of spin polarization in chGNRs crossing natural domain walls of the GdAu2 (revised supplementary Fig. S13, formerly Fig. S8). Here, one end of the GNR acts as a reference spin polarization, which allows to probe the opposite end and confirm that the magnetization of each end is driven by the spin direction of the GdAu2 region underneath. This is a complex experiment which was not well explained in the previous version. Consequently, we have included an arrow sketch (similar to the one in Fig. 1) in Supplementary Fig. S13.

Notwithstanding the above, we can give a bottom threshold for the interaction strength. While all presented data up to now are taken at 1.2 K, we were also able to measure a reliable spin polarization of the same approximate value (~8 %) at a temperature of 4.3 K. This data is shown in new Supp. Fig. S14 for a chGNR with a length of 26 precursor units. This means that exchange interaction between the chGNR and the GdAu2 is at least larger than the thermal energy corresponding to 4.3 K, which amounts to 0.4 meV.

Minor remarks:

1. The authors should include the theory counterpart of Fig. 2d.

Answer: Following the reviewer's recommendation, we have included the calculated LDoS of the theoretical MFH eigenstates in new supplemental Fig. S7. We find this more appropriate than adding a one to one comparison of theory and experiment in the main Fig. 2. The purpose of the comparison of theory and experiment in Fig2 or Fig 3 is not to show an exact match of them. The theory seeks to give a physical explanation of the experimental phenomenology, and associated energy scales. As such, the dI/dV maps in Fig 2D represent four different molecular orbitals, with distinct distributions at -40 mV and around Fermi level. Their assignment to confined states of the CB and VB has to be done by means of the quantitative Fourier analysis presented in revised supplemental Fig. S6. On the other hand, the theoretical LDoS simulations of the eigenstates shown in Fig 2E are an approximation of the molecular states in free standing GNRs (with $U=1$ eV though), which do not necessarily coincide with the experimental ones because some interactions that are present in the experimental set-up are overlooked. Consequently, we believe that the main text does not gain much clarity by comparing two objects which do not need to be identical in Fig 2.

However, the theoretical images in new Fig. S7 reinforce the interpretation of charge transfer to the GNR from the substrate and the assignment of the peaks at -40 mV and slightly below Fermi level to the onsets of the VB and CB respectively. It can be readily seen in Fig. S7 how the number of nodal points of the intensity distribution increases gradually from zero at -4.5 meV to 53 meV, as corresponds to discretized states of electron-pocket like CB. This is in very good agreement with the quantitative FFT analysis of the energy dependent dI/dV intensity in Fig. S6. The overall behaviour is best described with the addition of two electrons to the cGNR.

What are the missing ingredients in the theory that justify the different patterns of the molecular states along the edges? As discussed in the supplementary note 1 and in the context of main Fig 2, constant height scans with CO functionalized tips do not image the correct spatial distribution of the edge state due to the dominant contribution of the periodically

varying workfunction of the substrate (new supplemental Figure S5). In addition, CO tips do not survive long enough when scanning in constant current mode the highly reactive zig-zag segments. Therefore, we explore the electronic structure near Fermi level in Fig 2 with sharp metal tips in constant current mode. In this case, the effect of the moiré periodicity is still noticeable (revised supplemental Fig. S6B.2), but now the intrinsic confinement pattern of the different edge states can be discerned by means of the quantitative Fourier analysis presented in revised supplemental Fig. S6B (Note the modified colour scale of the revised Fig. 2D to highlight structure of the edge states). On top of these two combined periodicities, the constant current is known to be affected by topographic crosstalk, and as a result, the maps presented in Fig 2 do not portray an accurate representation of the vacuum LDOS distribution of the edge states.

2. Main text and the Methods section present two different justifications for reducing the magnitude of the Coulomb repulsion parameter U . Both make sense, but the two part of the text need to be brought in accordance.

Since this is an important concept, we mention the two reasons in the main text (lines 265-267) and, to avoid redundancy, remove this discussion from other parts of the article.

3. The introductory part needs a proofread for English; it also uses ambiguous terminology. Please improve.

The referee was right. We have rewritten it in a more direct style, using simpler vocabulary with clearer meaning for us, and corrected several prepositions.

- [1] M. Bazarnik, M. Abadia, J. Brede, M. Hermanowicz, E. Sierda, M. Elsebach, T. Hänke, and R. Wiesendanger, *Atomically Resolved Magnetic Structure of a Gd-Au Surface Alloy*, *Phys. Rev. B* **99**, 174419 (2019).
- [2] M. Corbetta, S. Ouazi, J. Borme, Y. Nahas, F. Donati, H. Oka, S. Wedekind, D. Sander, and J. Kirschner, *Magnetic Response and Spin Polarization of Bulk Cr Tips for In-Field Spin-Polarized Scanning Tunneling Microscopy*, *Jpn. J. Appl. Phys.* **51**, 030208 (2012).
- [3] J. Brede, N. Atodiresei, V. Caciuc, M. Bazarnik, A. Al-Zubi, S. Blügel, and R. Wiesendanger, *Long-Range Magnetic Coupling between Nanoscale Organic–Metal Hybrids Mediated by a Nanoskymion Lattice*, *Nature Nanotechnology* (2014).
- [4] L. Fernández, M. Blanco-Rey, M. Ilyn, L. Vitali, A. Magaña, A. Correa, P. Ohresser, J. E. Ortega, A. Ayuela, and F. Schiller, *Co Nanodot Arrays Grown on a GdAu₂ Template: Substrate/Nanodot Antiferromagnetic Coupling*, *Nano Lett.* **14**, 2977 (2014).
- [5] H. Söde, L. Talirz, O. Gröning, C. A. Pignedoli, R. Berger, X. Feng, K. Müllen, R. Fasel, and P. Ruffieux, *Electronic Band Dispersion of Graphene Nanoribbons via Fourier-Transformed Scanning Tunneling Spectroscopy*, *Phys. Rev. B* **91**, 045429 (2015).
- [6] T. König, G. H. Simon, H.-P. Rust, and M. Heyde, *Work Function Measurements of Thin Oxide Films on Metals—MgO on Ag(001)*, *The Journal of Physical Chemistry C* **113**, 11301 (2009).

Reviewers' Comments:

Reviewer #1:

Remarks to the Author:

The revisions have significantly improved the manuscript's quality, addressing the concerns I raised.

After reviewing the updated version, I believe the manuscript is now suitable for publication in Nature Communications.

Reviewer #2:

Remarks to the Author:

Authors have addressed all my concerns and I believe the revised manuscript is much clearer to readers on the novel detection technique of spins in carbon-based materials at the atomic scales. Therefore, I would recommend to publish in Nature Communications as it is.

Reviewer #3:

Remarks to the Author:

The authors have fully addressed my concerns, and I believe also remarks of the other referees. I'm ready to recommend the present version of this manuscript for publication in Nature Communications.